# Agent-Based Models Assisted by Supervised Learning: A Proposal for Model Specification

**Alejandro Platas-López \*,†**, **Alejandro Guerra-Hernández †**, **Marcela Quiroz-Castellanos †**
and **Nicandro Cruz-Ramírez †**

Instituto de Investigaciones en Inteligencia Artificial, Universidad Veracruzana, Xalapa 91000, Mexico
* Correspondence: alejandro.plataslo@anahuac.mx; Tel.: +52-228-221-5879
† Current address: Calle Paseo Lote II, Sección 2a No. 112, Nuevo Xalapa, Veracruz 91097, Mexico.

**Abstract:** Agent-based modeling (ABM) has become popular since it allows a direct representation of heterogeneous individual entities, their decisions, and their interactions, in a given space. With the increase in the amount of data in different domains, an opportunity to support the design, implementation, and analysis of these models, using Machine Learning techniques, has emerged. A vast and diverse literature evidences the interest and benefits of this symbiosis, but also exhibits the inadequacy of current specification standards, such as the Overview, Design concepts and Details (ODD) protocol, to cover such diversity and, in consequence, its lack of use. Given the relevance of standard specifications for the sake of reproducible ABMs, this paper proposes an extension to the ODD Protocol to provide a standardized description of the uses of Machine Learning (ML) in supporting agent-based modeling. The extension is based on categorization, a result of a broad, but integrated, review of the literature, considering the purpose of learning, the moment when the learning process is executed, the components of the model affected by learning, and the algorithms and data used in learning. The proposed extension of the ODD protocol allows orderly and transparent communication of ML workflows in ABM, facilitating its understanding and potential replication in other investigations. The presentation of a full-featured agent-based model of tax evasion illustrates the application of the proposed approach where the adoption of machine learning results in an error statistically significantly lower, with a *p*-value of 0.02 in the Wilcoxon signed-rank test. Furthermore, our analysis provides numerical estimates that reveal the strong impact of the penalty and tax rate on tax evasion. Future work considers other kinds of learning applications, e.g., the calibration of parameters and the analysis of the ABM results.

**Keywords:** agent-based modeling and simulation; machine learning; ODD protocol; tax evasion

## 1. Introduction

Modeling is an essential practice in science [1]. A scientific model is an abstract and simplified representation of a phenomenon of interest that makes its main features explicit and visible, and that can be used to understand the phenomenon, develop explanations, and generate predictions. From the natural to the social sciences, scientists have adopted different modeling alternatives. Among them, according to Gilbert [2], the agent-based modeling (ABM) approach is common, since it allows the development of models where individual entities and their interactions are directly represented, and it also makes it possible to model individual heterogeneity, specifying explicitly the rules of behavior of the agents and situating the agents in geographic or other space. This empowers modelers to represent multiple scales of analysis naturally, to ascertain macro- or societal-level structures from individual actions, and to employ various types of adaptation and learning. None of these advantages is easy to replicate with other modeling approaches, offering the modeler a lot of freedom to implement his or her model.

Despite this positive scenario, some specific disadvantages when implementing ABMs have been identified in the literature. They include the complexities involved in its design, the difficulty in calibrating its parameters, the laborious analysis and interpretation of its results, its high computational cost, and the programming skills required for its implementation. With the exception of the last two mentioned, the rest of these topics are related to various stages of ABM development, that is, the design, calibration, experimentation, validation, and analysis stages. Typically, these stages involve data processing before, during, or after the simulation itself. Machine Learning (ML) algorithms can be used in such processing, suggesting that ABM could take advantage of ML, given the greater availability of data, computational resources, and learning algorithms [3].

In recent years, more and more publications have considered the synergy between ML and ABM as a means to overcome some of the mentioned disadvantages. However, these efforts are still somewhat disorganized, as some researchers focus on assisting in model design [4–6], others focus on parameter calibration Chu et al. [7], Lamperti et al. [8], Zhang et al. [9], and still, others aim to find optimal behavior policies [10–12]. So, very often the contributions are are isolated and address particular problems rather than the emerging methodological issues that result from the integration of ML and ABM. Therefore, the main objective of this article was to propose a framework that allows the transparent description of the integration of both tools, in addition to providing a broad, but integrated, image of the current state of knowledge on ML–ABM interaction, emphasizing the different applications that exist to address specific problems in the different stages of development of agent-based models.

The rest of the article is structured as follows. Section 2 presents a literature review on the efforts made to synthesize the applications of ML in ABM, as well as computational models on tax evasion. Section 3 provides context by characterizing the criticisms of ABM reported in the literature, and identifying the kinds of ML algorithms considered to overcome these criticisms and the related issues. In Section 3 we also propose an extension to the ODD protocol [13,14] with the aim of overcoming the current limitations for a comprehensive and transparent description of the incorporation of learning in ABM. Section 4 presents a case study on tax evasion of the payroll tax in Mexico, which includes the ODD protocol in the terms proposed to describe the design of an ABM assisted by ML for the inclusion of a synthesized agent from existing data on the Mexican tax system. Section 5 presents the results of the implemented model, in which the versions of the model with and without learning are compared, highlighting the usefulness of synthesizing existing data to better capture the nature of the phenomenon under study. Finally, Section 6 offers conclusions.

## 2. Literature Review

This section summarizes the work that has been done to characterize the combination of machine learning tools and agent-based modeling. In the same way, since a full-featured agent-based model on tax evasion is presented as a case study, articles that address this phenomenon from the point of view of computer simulation are also presented.

### 2.1. Machine Learning in Agent-Based Modeling

There are previous works that have analyzed the incorporation of ML in agent-based models from a slightly broader perspective. Müller et al. [15] considered an extension to the ODD protocol, named ODD+D, to include human decision-making. It represents a correct step to complement the description of the agent-based models, which allows transparency and facilitates the understanding of the models. However, it does not refer to the different applications of ML in ABM that can be carried out, and, in any case, it refers only to the design of rules of behavior.

In a first contribution, Elbattah and Molloy [16] focused specifically on ML applications to assist in design, namely, data-driven feedback. However, they did not consider other

types of applications, such as online learning for behavior optimization or post-learning for calibration or analysis of results.

In a second contribution, Elbattah [17] considered three categories for applying ML in agent-based modeling, which they call conceptual modeling, runtime modeling, and output analysis. Although they considered other ways of including ML in ABM, they did not consider a posteriori offline learning for parameter calibration. Also, in neither of their two contributions did they indicate how a researcher could carry out the implementation and documentation of the incorporation of both tools.

Zhang et al. [18] distinguished four scenarios where ML could contribute to the ABM process, two of which were at the agent level (micro), and two at the model level (macro). As for the micro level, they differentiated two applications of ML, the first one being agent situational awareness learning, in which ML is commonly used to predict variables or agent-related behaviors. The second is based on the idea that as an agent lives in a changing environment, agents obtain more knowledge and experiences, which can be stored in the form of data. This data can be used to infer better decision-making models by applying appropriate ML techniques to intervene in the behaviors of this agent, allowing them to better achieve their goals.

At the model level, they mention two applications, the first is useful when the number of agents and parameters grows and the combinations of parameters grow exponentially, which generates a high computational cost, so ML is used to build the mapping structure (emulator) between the parameters and the emergency exit of the ABM. The second is to apply ML techniques to enhance the decision-making of the policymaker at the macro level. In these cases, the macro-level policymaker acts as a single "macro-agent".

Micro applications map to a priori offline learning and online learning, whereas macro applications map to post-hoc offline learning in their two tasks, validation-calibration, and analysis. However, although all possible tasks are taken into consideration, a methodology that allows researchers to carry out the transparent implementation is not proposed.

### 2.2. Tax Evasion

Tax evasion is an illegal and intentional activity taken by individuals to reduce their legally due tax obligations [19]. Although formal research on tax evasion began in the 1970s, with the seminal work of Allingham and Sandmo [20], who applied the economics of crime approach [21] to tax reporting behavior, scholars and practitioners continue to face the challenge of designing and implementing policies and incentives to mitigate tax evasion.

Taxes are essential to fund public expenditures. Therefore, tax evasion is not only a problem for the tax authorities but also a problem for society. Given that a country's investment in healthcare, education, national defense, social security, transportation, infrastructure, science, and technology mainly comes from public finances, tax evasion leads to mismatches in public goods [22]. So, the analysis of tax evasion can be used to determine the factors that affect the compliance rate and, ultimately, help the government achieve revenue targets. Daude et al. [23] found evidence of such factors in developing countries. They described two groups of factors: socioeconomic factors, such as age or education, and institutional determinants, like trust in government and satisfaction with the quality of public services.

Although early theoretical research provides a baseline for evaluating assumptions about tax evasion, the basic method of classical formulae, i.e., the use of utility functions and the assumptions of taxpayer homogeneity and rationality, insufficiently describe taxpayer behaviors observed in practice. To overcome this limitation in the study of tax evasion, an alternative, based on agent modeling and simulation, has been applied in this endeavor. So-called agent-based models (ABM) are designed to consider personal preferences and can accommodate a wider variety of internal and external variables to help explore a broader space for compliance results.

Mittone and Patelli [24] created a computational model of tax evasion that considers taxpayers' psychological motives and public sector goods. The authors assumed the exis-

tence of three classes of taxpayers. Each class has a unique utility function that describes its behavior. Taxpayer agents are assigned initially to one of the three categories but may change from one category to another during a model run by imitation of successful behavioral strategies. The model demonstrates that when audits are introduced the additional revenue raised increases the quantity of public goods that can be provided. This information is transmitted to the agents who, in turn, incorporate it into their utility calculations.

Davis et al. [25] analyzed the effect of social norms and enforcement on the dynamics of taxpayer compliance. They developed an analytical and a computational model to evaluate the movement between classes of compliant and non-compliant taxpayers. Their analysis suggested that the effect on compliance of changing enforcement levels depended on whether the taxpayer population was initially compliant or non-compliant. Compliant populations are insensitive to changes in enforcement policies until enforcement becomes sufficiently lax. In contrast, relatively non-compliant populations respond to increased enforcement by gradually increasing compliance. Then, when enforcement becomes sufficiently harsh, there is a sudden shift in equilibrium to very high levels of compliance. After the taxpayer population shifts from compliance to non-compliance, or vice versa, their models predicted that returning to the previous enforcement policy would not cause the population to return to its previous state. Their results also help to explain why taxpayer compliance varies across time and across geographic regions, even under similar enforcement regimes.

Hokamp [26] analyzed tax evasion dynamics and social interactions in heterogeneous populations with an agent-based model. They included aging to incorporate psychological findings with respect to social norms up-dating over a taxpayer's life cycle. The provision of public goods with an exponential utility function was also explored. Their findings supported the view that, when social norm updating is considered, age heterogeneity leads to fluctuations in income tax evasion.

## 3. Specification of Learning in Agent Based Modeling

Although agent-based modeling is a promising approach among several domains of science, some researchers have encountered drawbacks related to some of the stages of the agent-based modeling cycle. These disadvantages can be summarized as design problems [27–35], calibration–validation [28,29,34], interpretation of results and analysis [31,33–38], high computational cost [35,39–45], as well as necessary programming skills [30,37,46].

To overcome these limitations, some authors incorporated ML techniques at different stages of the development cycle of agent-based models. Broadly speaking, we can differentiate between applications with learning outside of the simulation, called offline learning, and learning during simulation, called online learning. In turn, offline learning can be classified into two aspects: a priori learning, in the ABM design stage, and a posteriori learning, applied in the calibration–validation and analysis phases. In the first case, learning uses existing data to instantiate model components. While in the second case, learning is applied to the data produced by the model, either to calibrate their parameters towards a state of validation of the results or to analyze the results by identifying which inputs produce which outputs and so develop a hypothesis of the simulation behavior. Figure 1 summarizes the different categories of ML applications in Agent-Based modeling. In what follows, some of the ML applications falling into the mentioned categories are briefly described.

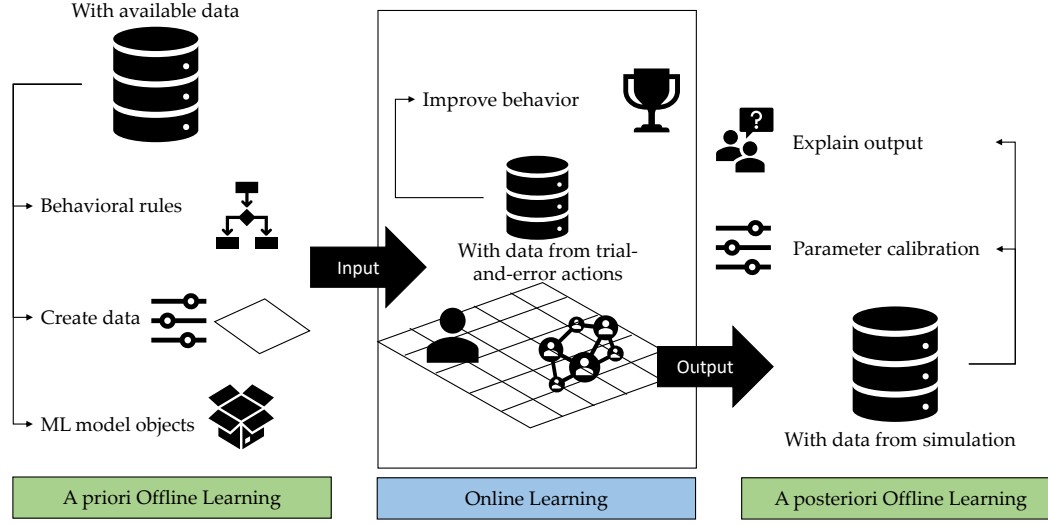

**Figure 1.** Categories of Machine Learning applications in Agent-Based modeling.

### 3.1. Design of Agent-Based Models Assisted by Machine Learning

The use of ML for design purposes implies the existence of data to carry out the learning. Among developments found in literature, the following three types of applications are distinguished: the first generates data to later use in the model [4,5,47], e.g., Saeedi [48] develops predictions of land use change with neural networks and, subsequently, implements the results as part of the environment; the second group of applications concerns learning the behavioral rules of the agents, so that the modeler can later implement them in the model [6,49,50]. It is important to note that in this type of implementation, the algorithms that are usually used are easy to interpret and allow the rules to be derived in a simple way, i.e., decision trees or Bayesian networks; and the third group of applications concerns learning an ML model from data, and later inserting that model in the ABM, so that it is available to agents, e.g., [51–53]. Unlike in previous applications, black box models, as artificial neural networks (ANNs), are preferred.

### 3.2. Validation and Analysis of Agent-Based Models Assisted by Machine Learning

The a posteriori offline learning applications that have been developed use data generated by the simulation. The use of ML for validation purposes consists in finding a subset of input parameters that can produce the desired output. This process is known as parameter calibration, and, in most cases, it has the purpose of performing validation of the model, i.e., ensuring that the model reproduces stylized facts of the real phenomenon. While for analysis purposes, the goal is to understand the complex relationships that occur within the model and to identify which inputs lead to which outputs.

A new approach for calibrating parameters of an agent-based model, using the Bayesian estimation technique to capture the fundamental features of economic fluctuations, is presented in Chu et al. [7]. Using Extreme Gradient Boosting, a tree-based algorithm, which sits under the supervised branch of ML, Lamperti et al. [8] proposes another approach to efficiently calibrating agent-based models to real data. Their proposed method, "learns" a fast surrogate meta-model using a limited number of ABM evaluations and approximates the nonlinear relationship between ABM inputs (initial conditions and parameters) and outputs. A similar approach is presented by Zhang et al. [9] with another ensemble algorithm called CatBoost.

In an attempt to make the problem of parameter tuning more tractable, van der Hoog [54] emulated the input/output function of the entire ABM by a less complex and more tractable Deep Neural Network. This is useful for robustness and parameter sensitivity analysis, since it

allows a much larger exploration of the parameter space. Other approaches involve the use of heuristics [55] and metaheuristics [56] for parameter optimization.

An ABM calibration approach proposed by Ye et al. [57], established a link between agent micro-behavioral parameters and systemic macro-observations. To compress the agent state space, principal component analysis was introduced to avoid the high dimensions of a macrostate transfer equation. Another approach was proposed by Kim et al. [58], in which automatic calibration was done with both dynamic and heterogeneous calibrations, using Hidden Markov Model, Variational Autoencoder, and Gaussian Process Regression.

In order to analyze the spread of *Cutaneous Leishmaniasis* in central Iran, Rajabi et al. [59] proposed an ABM to simulate the dynamics of spread based on a Geographic Automata System, the results of which were analyzed by means of Bayesian modeling. In a similar approach, Hayashi et al. [60] used Decision Trees in order to achieve comprehensive behavior prediction in the case of customer churning in a subscription-based business. With a similar objective, R. Vahdati et al. [61] also used Decision Trees to understand how factors, such as climate, ecology, human behavior, and population dynamics, interacted to affect human survival and dispersal from Africa in the Late Pleistocene age. As can be seen, these approximations use easy-to-interpret or white-box algorithms.

However, the use of black box algorithms is more frequent, e.g., Ozik et al. [62] integrated random forest or artificial neural networks to dynamically and efficiently characterize the parameter space of a large and complex agent-based infectious disease model. Similarly, Chen et al. [63] used random forest regression to evaluate the correlation between parameters and outputs and to then build a metamodel by a neural network to predict the simulation outputs from the parameters. Random forests were also implemented to make inferences from a detailed model representing past human–environment interactions [64]. Garg et al. [65] identified model parameters that were most influential to ABM outputs. Edali and Yücel [66] used the parameters to discover the relationship between inputs and outputs of agent-based models. Gursoy and Badur [67] provided quick predictions on the number of emigrants to understand the effect sizes of individual parameters.

Regarding proposals for analysis through ANN, Xu et al. [68] revealed the learning process and heterogeneity of agents in the urban expansion of Auckland, New Zealand. To predict the time evolution of an ABM, focusing on the clinical condition of sepsis, Xu et al. [68] trained an ANN as a surrogate system. Equally, Xiao and Liu [69] trained an ANN to study the effects of different combinations of control measures for COVID-19. For urban decision-making, Zhang et al. [70] used a convolutional neural network to achieve real-time prediction of an agent-based model. Other proposals considered the use of Support Vector machines and support vector regressions [71], causal discovery [72], reconstructability analysis [73] and the Louvain algorithm [74], an unsupervised algorithm to cluster structures in social networks.

### 3.3. Online Learning in ABM Assisted by Machine Learning

Online learning makes use of data generated during the simulation. Its purpose is to generate optimal behavior for agents by allowing them to explore the environment through trial and error. Due to the type of learning task, most of the proposals opt for Reinforcement Learning (RL) algorithms. Many works in the literature use Temporal Difference, and, specifically, the Q-learning algorithm [10–12,75–78]. Other Temporal Difference algorithms are also used [79,80]. Liang et al. [81] proposed the Deep Deterministic Policy Gradient Algorithm and Deep Q-Networks were proposed by Sert et al. [82].

A few exceptions adopted supervised learning algorithms. Jäger [83] introduced a framework that, although closely related to RL, used ANN to enable the agents to learn rules. An expectation model of inflation, based on ANN, was introduced by Salle [84], wherein the weights were randomly initialized so that, at least at the beginning, agents' mental models and the resulting expectations differed. As new observations became available, agents' ANN were trained. In the energy market domain, Dehghanpour et al. [85] proposed that each company agent develops a private probabilistic model of the market

using dynamic Bayesian networks. Sparse Bayesian Learning, an online learning algorithm, was used for training the model to infer the future state of the market and to estimate an optimal bidding function.

Although most of the applications have the objective of finding the optimal rule of behavior and using this rule as a hypothesis of the phenomenon, there are some approaches that try to assist the modeler in the design of the ABM by returning the found rule to be implemented. A novel path to agent policy development was examined by Norman et al. [86], in which modelers did not manually craft the policies, but allowed them to emerge through the application of RL within a game engine environment. Fuller et al. [87] proposed the use of RL to enable agents to define their required interaction rules. An approach in which agents created their own optimal strategy and the designer could interpret it was given by Cummings and Crooks [88].

### 3.4. The Overview, Design Concepts and Details Protocol

The Overview, Design concepts, and Details (ODD) protocol [13,14], is a standardized document providing a consistent, logical, and readable account of the structure and dynamics of ABMs. It consists of seven elements, as shown in Figure 2. These elements are grouped into three levels of description, ranging from general to specific.

**Overview.** A general description of the model, including its purpose and its basic components, the agents and variables describing them and the environment, and the scales used in the model, e.g., time and space, as well as an overview of the processes and their scheduling.

**Design concepts.** A brief description of the basic principles underlying the model's design, e.g., rationality, emergence, adaptation, learning, etc.

**Details.** Full definitions of the involved sub-models.

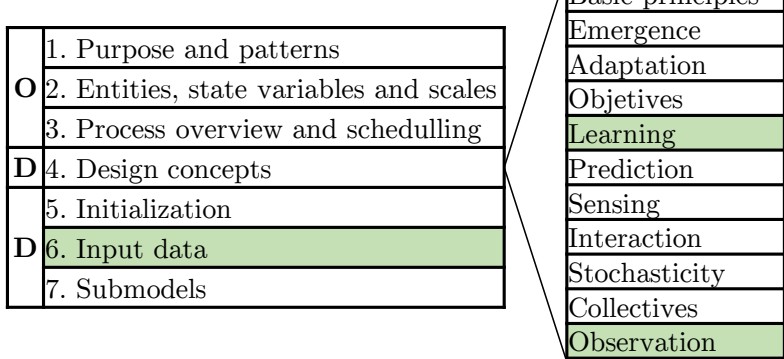

**Figure 2.** Structure of model descriptions following the ODD protocol. Elements, where an update is proposed, are highlighted. Adapted from [14].

The inclusion of ML in ABM occurred gradually, si the original ODD protocol [89] did not consider learning among the design elements. In the first update to the protocol [13], the relevance of learning was recognized, including it as a design concept, but adopting only the online modality. Instead of enhancing the specification of learning as a design element, the second update to the ODD protocol [14] focused on other aspects, such as clarity, replication, and structural realism. Indeed, in its current state, the only questions about learning in the ODD protocol are the following: "Do many individuals or agents (but also organizations and institutions) change their adaptive traits over time as a consequence of their experience? If so, how?".

So, the learning section in the current ODD protocol is intended to describe online learning tasks only, leaving out the possibility of documenting offline learning applications,

both before and after simulation, such as those referenced previously. An extension to the ODD protocol enabling the description of all these learning tasks seems necessary.

*3.5. Proposal*

We propose extending the ODD questions associated with learning as a design element covering:

1. **What is the purpose of learning?** Following the approach of the current ODD Protocol, as stated by Grimm et al. [13], a possible purpose of learning is to improve the performance of the agents during simulation based on their experience. Other equally valid purposes, as discussed above, include the following: generating data for simulations, inducing the rules defining the behavior of the agents from existing data, calibrating parameters of the model, and analyzing the results of the simulations.

2. **When is learning performed?** According to its purpose, learning can take place at different moments. In the current ODD Protocol, learning is expected to occur during the simulation. It is also possible to learn before the simulation, to obtain models to generate synthetic data to be used during simulation, or the rules of the agents included in the ABM. It is also possible for the learning to occure after the simulation to calibrate parameters or to analyze the data produced by the simulation.

3. **What components of the ABM are affected by learning?** In the current ODD Protocol, agents are the only components performing, and being affected by, learning. However, it is also possible that the environment exploits the use of ML methods, e.g., consuming synthetic data generated by learned models. It is also possible to learn the values of the parameters of the ABM to obtain some desired behavior of the whole system.

4. **How is learning computed?** Finally, the type of algorithm used and its input data must be considered. Online learning requires incremental learning algorithms, such as those included in, but not limited to, RL. Supervised techniques can be used for offline learning, both before and after simulation. Evolutionary approaches are well suited for calibrating parameters. The preference for explainable methods versus black box techniques is also related to the purpose of learning.

Some considerations must be addressed in other sections of the ODD protocol to obtain a more comprehensive description of the learning process. They are described as follows.

A priori ML implies the use of existing data related to the phenomenon to be simulated. The use of such data must be included in the details section of the ODD Protocol, specifically in the element called "Input data", in which the following question is answered:

---

**6 Input data**

**Current**: Does the model use input from external sources, such as data files or other models, to represent processes that change over time?

- - - - - - - - - - - - - - - - - - - - - - - - - - - - - - - - - - - - - - - - - - - - - -

**Proposal**: Does the model use data from external sources (data files, or other models) to represent **some element in the model**? **Is the data processed with machine learning?**

---

A posteriori ML implies the use of data resulting from the simulation. The use of such data must be included in the design concepts section of the ODD Protocol, and, specifically, in the element called "Observation", in which the following question is answered:

> ### 5.11 Observation
>
> **Current**: What data are collected from the ABM for testing, understanding, and analyzing it, and how and when are they collected? Are all output data freely used, or are only certain data sampled and used, to imitate what can be observed in an empirical study?
>
> - - - - - - - - - - - - - - - - - - - - - - - - - - - - - - - - - - - - - - - - - - - - - - -
>
> **Proposal**: What data are collected from the ABM for **calibrating**, testing, understanding, and analyzing it, and how and when are they collected? Are all output data freely used, or are only certain data sampled and used, to imitate what can be observed in an empirical study? **Is the data processed with machine learning?**

The questions rewritten in this way link the source of the data with the learning section in which the learning task to be performed is described. Finally, a complete description of the ML model must be provided in the ODD protocol sub-models, including data preparation, modeling, and evaluation.

## 4. Case Study: The Payroll Tax Evasion Model

This section describes in detail an ABM in which the design is assisted by ML. This specification adheres to the extension proposal to the ODD protocol made in the previous section. It is important to highlight that the case study is a full-featured model, in which the learning application has favorable repercussions for the analysis of fiscal policy. As it is an application of ABM design, assisted by ML, the aspects of the ODD where the improvements are made are in the learning design concept, the input data, and the sub-model, Section 2, where the ML algorithm used is described in detail.

The purpose of this model is to advance the state of the art in tax evasion analysis, presenting an alternative based on agent simulation and machine learning. Where certain changes in taxpayer behavior can be mined from data and applied during the simulation, which otherwise would not be detected by the model. Evasion of payroll tax in Mexico is modeled as a case study to exemplify the benefits of the combination of ML and ABMs. Each Mexican state has autonomy over the way in which payroll tax is collected. Therefore, to model these different fiscal scenarios and their effects, an explicit representation of the space is made through a Geographic Information System with hexagonal tessellation.

For the sake of reproducibility, the model is described in the following sections using the ODD protocol [13,14].

### 4.1. Overview

Purpose. Analyze the effect of institutional determinants on payroll tax evasion in Mexico.

Entities, state variables, and scales.

- Agents: Employers and tax authority.
- Environment: A Mexican state-level representation of the payroll tax system.
- Scales: Time is represented in discrete periods, each step representing a month, which corresponds to the tax collection period according to the current legislation. In order to have a margin of error less than 5% with a confidence interval of 99% in the selected sample, each employee agent in the model represents 2000 employers in the 2019 Mexican labor market.
- State variables: The attributes that characterize each agent are shown in Table 1 along with the method for initializing each variable.

Process overview and scheduling.

1. Employers decide in which market to, formally or informally, query a machine learning model.

2.  Informal employers are full evaders. Formal employers calculate the amount of taxes to report. If they calculate so as to report zero taxes, they also become full evaders. If they calculate to report all the tax, they become full taxpayers. In any other case, they become partial evaders.
3.  Tax authority collects the declared amount of taxes.
4.  The tax authority conducts audits on a random basis. If the audit is successful, then partial and full evaders must pay the evaded amount and a penalty for the undeclared amount.
5.  Every 12 months employers increase their age. With some probability, in each period, employers can die. If this happens, they are replaced by another employer with the same characteristics, except for age.

*4.2. Design Concepts*

Basic principles. Voluntarily declared income rises with increasing individual income, penalty rate, and audit probability, and decreases with increasing tax rate, as proposed in the neoclassical theory of evasion by Allingham and Sandmo [20], and also includes exponential utility and public goods provision, described by Hokamp [26]. The distribution of production among employers follows a power law in which there is a marked inequality in the distribution, characterized by a small percentage of people holding the most resources [90]. This characterizes a capitalist economy, such as that in Mexico. Mortality of employers follows a Weibull distribution [91]. This function is adjusted for the case of Mexico, where life expectancy at birth is seventy-five years [92].

Emergence. The extension of tax evasion is an emerging result of the adaptive behavior of employers. These results are expected to vary in complex ways when the characteristics of individuals or their environment change. Other outcomes, such as the distribution of production and age, are more strictly imposed by rules and, therefore, less dependent on what individuals do. However, these results are important for the validation of the model.

Adaptation. Employers can change their alignment in the formal sector. To make that decision, they query a previously trained machine learning model. With this trait, agents could become full evaders, and, therefore, avoid paying taxes. The tax authority does not have adaptation mechanisms.

Objectives. The employer's goal is to increase his utility by paying the least taxes. The objective of the tax authority is to increase tax collection.

**Learning**. With the extension proposal for learning inclusion, the following questions are answered. *What is the purpose of learning?* The purpose of learning is to deduce the rules defining the behavior of the agents from existing data. *When is learning performed?* Learning is performed before the simulation to obtain models for generating the rules of the agents included in the ABM. *What components of the ABM are affected by learning?* The environment is affected by encapsulating, in a learned model, the data of the fiscal system and the quality of public goods. *How is learning computed?* Learning is computed by a supervised algorithm.

Prediction. The employers' learning is an offline process. During the simulation, employers sense their current conditions and act accordingly. Employers do not have an internal model to estimate their future conditions or the consequences of their decisions.

Sensing. Employers consider the size of their company, education, sector of occupation, state, size of the locality, age, and income as internal states, and they perceive tax rate, and level of insecurity as environmental variables. If an audit is successful, the tax authority can collect undeclared tax from employers.

Interaction. Interactions between employers and the tax authority are direct when, with a certain probability, an audit is carried out.

Stochasticity. The audit process is assumed to be random, with both the probability of it being carried out and its probability of success following a uniform distribution. The process of initialization of the employers' income is also considered random, following a power law. The probability of death of employers is also random following a Weibull

survival distribution. In the first process, stochasticity is used to make events occur with a specific frequency. In the last two processes, stochasticity is used to reproduce the variability in processes for which it is not important to model the real causes of the variability.

Collectives. Employers are grouped into three types of tax behavior: full evaders, partial evaders, and full taxpayers. This collective is a definition of the model in which the set of employers with certain properties about the amount of taxes paid is defined as a separate kind of employer with its own variables and traits.

Observation. At the end of each run, data is collected on the extent of tax evasion, the amount of tax collected, and the number of full evader employers.

*4.3. Details*

Initialization. At the time $t = 0$, of every simulation run, $N = 1337$ employers were initialized and distributed in each state according to the input data. A total of 32 auditors were also initialized, representing each state tax authority. Some state variables were initialized by a submodel, either deterministic or random, while others were based on data, as shown in Table 1.

**Table 1.** State variables and method for initialization. In parentheses, is the name of the attribute in the National Institute of Statistics and Geography (INEGI) dataset [93] of state variables.

| Agent | Attributes | Type | Initialization | Value |
|---|---|---|---|---|
| Auditor | penalty-collected | Float | Deterministic | 0 |
| | tax-collected | Float | | 0 |
| | my-employers | AgSet | | Submodel 5 |
| | ent-auditor | Int | Random | $[1, 32]$ |
| Employer | business-size (ambito2) | Int | Database | $\{0, 2, 3, 4, 5, 8\}$ |
| | education (anios_esc) | Int | | $[0, 20]$ |
| | economic-activity (c_ocu11c) | Int | | $[1, 10]$ |
| | age (eda) | Int | | $[17, 98]$ |
| | mexican-state (ent) | Int | | $[1, 32]$ |
| | income (ing7c) | Int | | $[1, 7]$ |
| | formal-or-informal (mh_col) | Int | | $[0, 1]$ |
| | size-of-region (t_loc) | Int | | $[1, 4]$ |
| | corruption | Float | | $(0, 1)$ |
| | insecurity | Float | | $(0, 1)$ |
| | tax | Float | | $(0, 1)$ |
| | audit? | Bool | Deterministic | false |
| | audited? | Bool | | false |
| | type-of-taxpayer | Int | | 2 |
| | declared-tax | Float | | 0 |
| | payroll | Float | | Submodel 11 |
| | payroll * | Float | | Submodel 12 |
| | risk-aversion-$\rho$ | Float | | Submodel 16 |
| | undeclared-payroll | Float | | 0 |
| | undeclared-tax | Float | | 0 |
| | $\alpha$-s | Float | | 0.05 |
| | $\delta$ | Float | | $-0.1$ |
| | prob-formal | Float | Random | $(0, 1)$ |
| | production | Float | | Submodel 10 |

In the same way, input parameters of simulation were adopted from literature, databases or through experimentation. Table 2 shows the initial values of the baseline model. In Mexico, the penalty rate varies in some states between 55 and 75%, and in others between 75 and 100%, so an initial value of 75% was adopted. The variation in tax rates, perception of insecurity, and corruption started at zero, that is, the value indicated in local tax legislation and in INEGI was taken. According to Bonet and Rueda [94], the effectiveness of tax collection in Mexico is approximately 0.7. For the decision threshold, 0.5 was used, since, according to [95], the threshold of 0.5 is commonly used to estimate the

probability for a set with two classes. For the audit probability and the effectiveness of the audits, values were found through experimentation to guarantee outputs similar to the dynamics of the Mexican tax system, which is small but efficient. The user interface also had other input parameters, one of them allowing switching the machine learning model "on" and "off".

**Table 2.** Input parameter initialization of baseline model.

| Parameter | Description | Value | Initialization |
|---|---|---|---|
| $\pi$ | Penalty rate | 0.75 | Database |
| $\alpha$ | Audit probability | 0.05 | Experimentation |
| $\epsilon_{AP}$ | Effectiveness of audit process | 0.75 | Experimentation |
| $\epsilon_{TC}$ | Effectiveness of tax collection | 0.70 | Literature [94] |
| $\Delta\theta$ | Variation in tax rate | 0.00 | Database |
| $\Delta PI$ | Variation in perceived insecurity | 0.00 | Database |
| $\Delta PC$ | Variation in perceived corruption | 0.00 | Database |
| $\tau$ | Threshold for formal or informal sector choice | 0.50 | Literature [95] |

**Input data**. *Does the model use data from external sources (data files, or other models) to represent some element in the model?* Yes, the model used external input data to learn the Mexican tax system, that is, the National Survey of Occupation and Employment (ENOE), and the National Survey of Quality and Government Impact (ENCIG), as well as the tax laws of the different states where the payroll tax rate was specified. The period considered for the 3 sources of information was from 2011 to 2019.

ENOE is a quarterly survey, but just the third quarter was used as a reference for annual information. Among other data, it offers sociodemographic variables on the characteristics of the employers. From these variables, it could be determined whether an employer was in the formal or informal sector. ENCIG is a biannual survey in which people are asked about the top 3 (three) problems they believe exist in their state. The main problems among all the states were insecurity, corruption, and unemployment. Insecurity and corruption were summarized by state and interpolated to get annual information. The resulting data was joined to the ENOE data set along with the tax data. The dataset collected in this way gave a matrix of size 71,833 × 10, where the proportion of formal employers was 60.57%.

*Is the data processed with machine learning?* Yes, from the resulting pre-processed ENOE data, sampling was performed using the local pivot method with the algorithm offered by the R package "SamplingBigData" [96], which effectively generated a balanced sample data set. Selected attributes to generate the balanced sample were the state, the employer's classification in the formal or informal sector, and the size of the employer's economic unit. This ensured that the employers in the model reflected the actual proportions of the Mexican labor market. The size of the selected sample corresponded to a scale of 1 to 2000. Figure 3 shows the ML integration process.

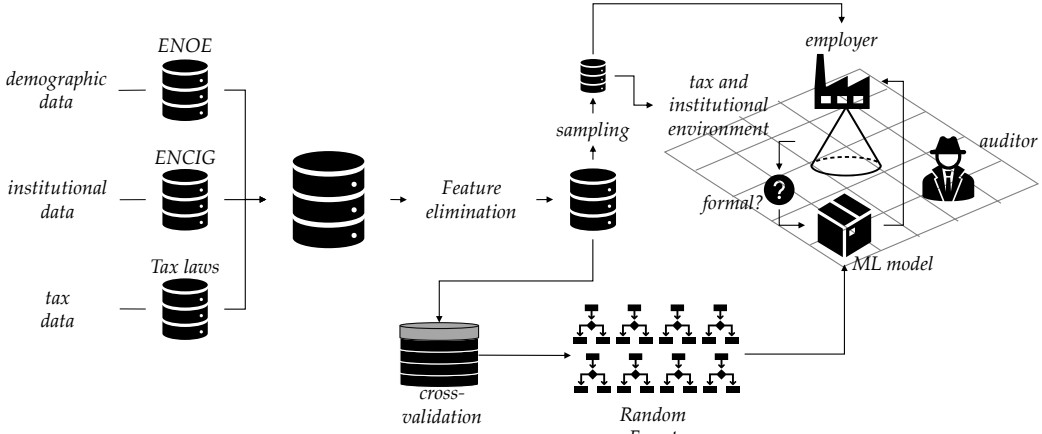

**Figure 3.** Diagram of integration of ML in the Payroll Tax Evasion Model.

**Submodels.**

1. After performing a pre-processing of the database, including a manual selection and a recursive feature elimination with resampling algorithm [97] available in R package "caret" [98], it was determined that the main variables that determined the employer's sector were the size of the business (ambito2), education (anios_esc), economic activity (c_ocu11c), state (ent), size of the region (t_loc), and age (eda).

2. A fast implementation of Random Forest in the R package "ranger" [99] was chosen to learn from data because it provides fast model fitting and evaluation, is robust to outliers, can deal with simple linear and complicated nonlinear associations, and produces competitive prediction accuracy [100]. To tune the hyperparameters and evaluate the performance of the model, cross-validation with $k = 10$ folds was carried out. The final hyperparameter values were $mtry = 20$, $ntrees = 100$, and $nodesize = 1$. That setting provided an accuracy of 83.79%, which was considered good to avoid overfitting. The trained model was available to employers during the simulation.

3. A Geographical Information System layer was loaded. Each polygon was a hexagonal tessellation of the corresponding Mexican state.

4. $N = 1337$ employers were generated and initialized with information from the database and moved to their corresponding state.

5. Auditors were generated and located in their assigned state.

6. The a priori learned random forest model was loaded.

7. Pareto-law values with the following distribution function were generated:

$$f(x) \sim x^{-1-\gamma}$$

8. Where $\gamma$ was known as the Pareto exponent and estimated to be $\approx 3/2$ to characterize a capitalist economy [90].

9. $x$ are the values generated by a normal distribution function with a mean of 2 and a standard deviation of 0.2 for informal employers and 0.3 for formal ones.

10. To assign a fixed monthly production value to each employer. Generated power law values were multiplied by 23 in the case of informal employers and 50 for the formal. Those quantities generated a perfectly mixed Pareto distribution according to the basic principles and preserved the participation of the informal economy in Mexican Gross Domestic Product (GDP) [101].

11. For simplicity, it was assumed that each employer allocated 30 percent of the value of production to payroll W. The share of wages in Mexican GDP was between 30 and 40% [102].

12. At the beginning of the simulation, it was assumed that non-informal employers declared all the tax, i.e., declared payroll $W^* = W$.

13. At the beginning declared tax $X^*$ by each employer was equal to the declared payroll $W^*$ multiplied by the tax rate $\theta$ in the employer's state.
14. Every 12 periods (months) employers increased their age, and they consulted the learned model to decide whether to opt for the formal or informal market, taking their internal attributes and perceived insecurity as a reference.
15. Informal employers did not declare taxes.
16. By social norm [26], employers modified their risk aversion $\rho$ according to their age, as follows:
$$\rho \sim \begin{cases} U(0.0, 0.25) & \text{if} \quad \text{age} \leq 34 \\ U(0.25, 0.5) & \text{if} \quad 34 < \text{age} \leq 51 \\ U(0.5, 0.75) & \text{if} \quad 51 < \text{age} \leq 67 \\ U(0.75, 1.0) & \text{if} \quad \text{age} \geq 67 \end{cases}$$

17. Let $\beta$ the perceived public goods efficiency, and $\pi$ the penalty rate.
18. Let $\epsilon_{AP}$ and $\epsilon_{TC}$ the effectiveness of audit process and tax collection, respectively.
19. Let $\alpha$ the true audit probability and $\alpha_S$ the subjective audit probability known to the employer.
20. Let $\delta = 0.1$, the updating parameter for $\alpha_S$.
21. If an employer was audited in a specific period, subjective audit probability became 1.
22. In each period (if not audited again) $\alpha_S$ decreased in $\delta$ amount until $\alpha_S = \alpha$.
23. In each period, employers calculated the amount of taxes to declare voluntarily $X^*$, applying the expected utility maximization procedure adopted by Allingham and Sandmo [20]. Let lower bound be:
$$\alpha_S > \frac{1}{1 + \left( \frac{(1-\beta(1-\epsilon_{AP}))\pi}{(1-\beta(1-\epsilon_{TC}))\theta} - 1 \right) e^{\rho(1-\beta(1-\epsilon_{AP}))(\pi W)}}$$

24. And the upper bound be:
$$\alpha_S < \frac{1}{1 + \left( \frac{(1-\beta(1-\epsilon_{AP}))\pi}{(1-\beta(1-\epsilon_{TC}))\theta} - 1 \right) e^{\rho(1-\beta(1-\epsilon_{AP}))}}$$

25. If the subjective audit probability $\alpha_S$ exceeded the upper limit in submodel 22, the employer became fully tax compliant, that is, $X^* = W\theta$, and when $\alpha_S$ fell below the lower bound in submodel 21, the employer fully evaded, that is $X^* = 0$.
26. For $\alpha_S$ in the range for an inner solution, the employer voluntarily declared:
$$X^* = W - \frac{\ln\left( \frac{(1-\alpha_S)(1-\beta(1-\epsilon_{TC}))\theta}{\alpha_S((1-\beta(1-\epsilon_{AP}))\pi - (1-\beta(1-\epsilon_{TC}))\theta))} \right)}{\rho\pi(1 - \beta(1 - \epsilon_{AP}))}$$

27. The tax authority collected payroll taxes that employers voluntarily declared.
28. The tax authority carried out audits with a random probability of $\alpha$ and a level of effectiveness $\epsilon_{AP}$.
29. If an evader was detected the undeclared tax was collected and a penalty rate $\pi$ applied over the undeclared tax.
30. In each period, employers had a probability of dying, following a Weibull quantile derivation function:
$$Q(p) = \lambda \left[ \frac{1}{1-p} \right]$$

31. Where $\lambda = 0.019$ and $k = 0.479$ are the scale and shape parameters, respectively.

32. It was assumed that, when an employer died, someone else took their place with the same attributes, except for age, which was generated according to:

$$eda = \lfloor X \rfloor$$
$$X \sim N(\mu, \sigma^2) \sim N(37, 6)$$

33. At each time $t$, the observed output Extent of Tax Evasion (ETE) was calculated as follows:

$$ETE_t = 1 - \frac{\sum_{i=1}^{N} W^*}{\sum_{i=1}^{N} W}$$

## 5. Results

The model was implemented in Netlogo [103]. Figure 4 shows part of the resulting GUI that allowed the initialization of global parameters and provided a view of the agents in a grid environment. The color of the hexagons represented the tax collected by the entity. Employing agents were colored blue, red, or cyan, depending on whether they were tax compliant, evading, or partially evading, respectively. The monitors on the right side show the outputs produced in the model with the given parameter settings.

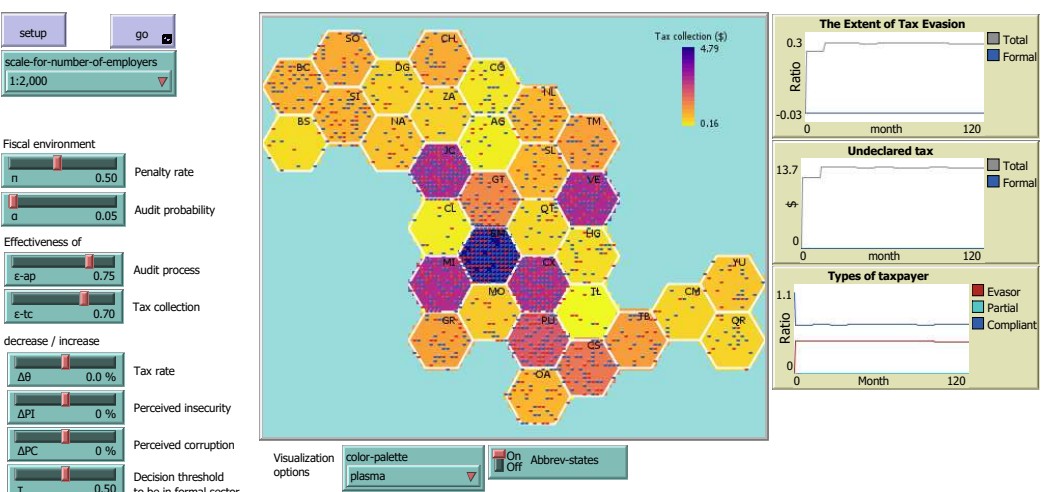

**Figure 4.** Graphical user interface of the model.

Although the demonstrated simulation model is useful, for "what-if" studies that explore different scenarios, policy analysis requires validated descriptive simulation models [104]. In this section, a validation of the results of the baseline model was carried out in micro and macro terms. Validation consists of replicating the statistical properties of the observed data. After validation, a comparison was performed between the simulation outputs of the models with and without machine learning. Assessment was made in terms of the extent of tax evasion, collected taxes, and the proportion of employers that fully evaded.

### 5.1. Validation

Empirical validation of agent-based models in economics has undergone significant development [105]. Among the different validation proposals, the indirect approach is one of the most used and the first evaluation that an ABM must satisfy. The core of indirect validation is that micro and macro variables can generate stylized facts or statistical properties that the modeler can compare with those obtained from empirical analysis of the corresponding real-world data set [105,106]. An indirect approach was used to carry out the validation since there were output variables that should follow stylized facts, as well as real data to carry out the comparison.

At the micro level, validation consists of verifying the existence of some stylized facts in economics. As mentioned previously, one of the basic principles lies in the assumption that, in a capitalist economy, the distribution of production follows a power law. This distribution is also transferred to the distribution of the payroll, assuming that greater production demands a greater amount of wages. Figure 5 shows that submodels 5 to 9, which adhered to the basic principle of the Pareto distribution, generated a power law in the payroll distribution, i.e., a positive bias in the distribution. Figure 6 shows the positive bias of 25 independent runs, which was measured with the skewness, a measure of the asymmetry of the probability distribution of a real-valued random variable about its mean [107].

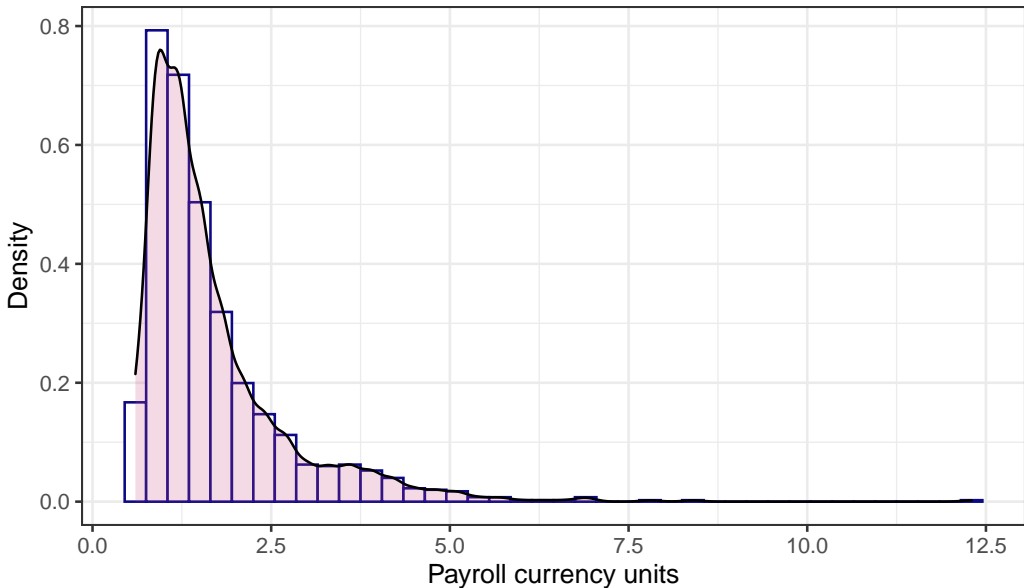

**Figure 5.** Distribution of payroll follows a power law.

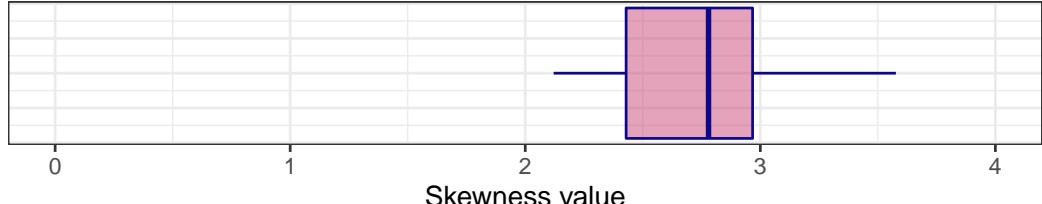

**Figure 6.** Skewness values obtained over 25 independent runs of payroll distribution. It was considered that values greater than 1 corresponded to highly positively skewed distributions.

At the macro level, a comparison was made between the amounts of tax collection obtained in the simulation and those reflected in the official statistics by state. The data on tax collection that had simulation as a source, were divided into simulations with machine learning "on" and "off". For the simulation data, the values corresponded to the arithmetic mean of 25 independent runs of the values simulated last year. The values of the real source corresponded to those reported by INEGI in 2019 [108]. Since the "real" data source contained quantities much larger than those produced in the model, all the data were centered and scaled to make the data comparable. Figure 7 shows the distribution of payroll tax collection by the data source. Visually, the distribution of the payroll tax collection approximated the data found in the statistics.

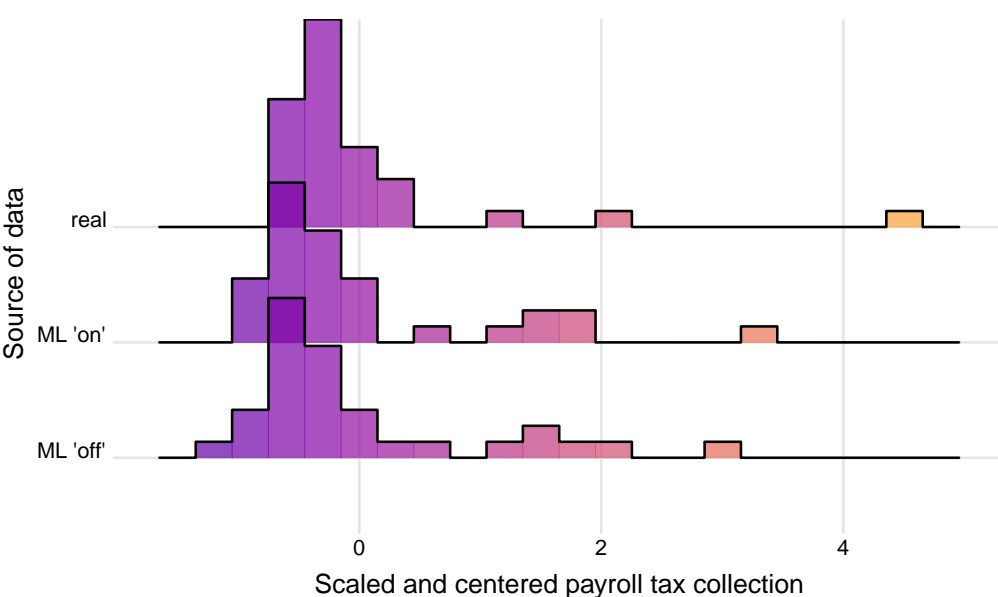

**Figure 7.** Distribution of scaled and centered payroll tax collection by the source of data.

The values of the state collection of the payroll tax from the statistics for the period 2010 to 2019, and from the arithmetic mean of 25 independent runs of the simulated last year with and without machine learning, were ranked from least to highest. The monotonic relationship between these data sources was measured with Spearman's correlation coefficient. In politics, absolute values greater than, or equal to, 0.6 are considered a strong correlation [109]. Table 3 shows the Spearman correlation coefficients between these data. A strong correlation was obtained for all the years considered. Both micro and macro validations demonstrated that the baseline model was a promising tool for analyzing payroll tax evasion.

**Table 3.** Input parameter initialization of baseline model.

| Model | Year of Real Source | | | | | | | | | |
|---|---|---|---|---|---|---|---|---|---|---|
| | 2010 | 2011 | 2012 | 2013 | 2014 | 2015 | 2016 | 2017 | 2018 | 2019 |
| ML 'off' | 0.73 | 0.75 | 0.68 | 0.68 | 0.64 | 0.61 | 0.61 | 0.60 | 0.60 | 0.61 |
| ML 'on' | 0.71 | 0.73 | 0.68 | 0.68 | 0.65 | 0.62 | 0.60 | 0.60 | 0.60 | 0.61 |

Finally, the Root Mean Squared Error (RMSE) was computed to validate the results of the models with and without machine learning, when compared to the actual values. Figure 8 summarizes by Mexican state and model, the error between the predicted annual tax collection in 10 simulated years and the actual collection in the period 2010 to 2019. The RMSE in the model with machine learning was lower in 19 of the 32 states. To determine if these differences were significant, a test was performed to compare a location measure, since the error of both samples, for the treatments without and with learning, were not normal. The Wilcoxon signed-rank test was used. The alternative hypothesis was that the median RMSE without learning was greater than the median RMSE after learning. The $p$-value of the test was 0.02166, which was less than the $\alpha$ significance level of 0.05. We could conclude that the median RMSE before ML was significantly higher than the median RMSE after ML.

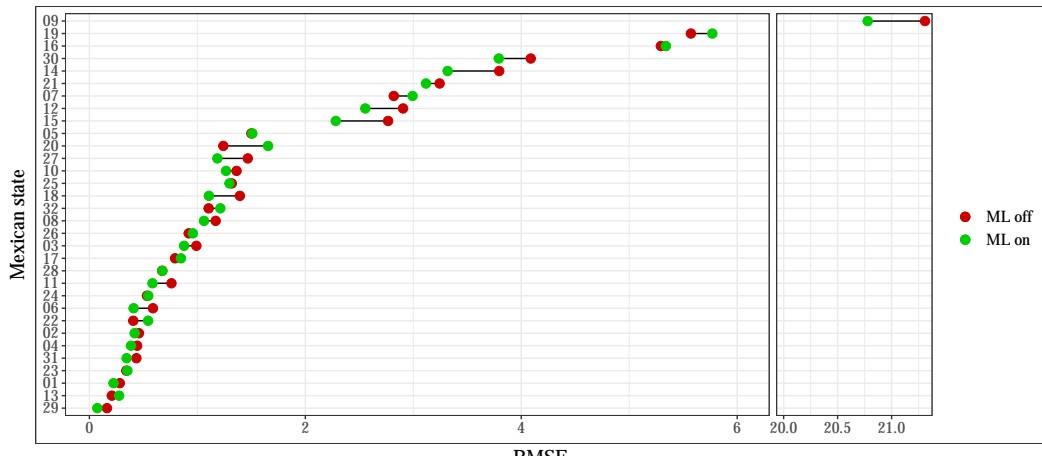

**Figure 8.** Root Mean Square Error (RMSE) of predicted and actual payroll tax collection in 10 simulated years by state with machine learning "on" and "off". The x-axis break was done to effectively utilize plotting space and deal with outliers [110].

### 5.2. The Effect of Machine Learning in Simulation

Figure 9 shows the variations in the outputs of the three observed variables when ML was turned "on" or "off" and changes in the perceived corruption or insecurity were introduced. The value observed in the data corresponded to the coordinate 0.0. Negative values represented a decrease in the perceived value in the respective institutional factor, i.e., less corruption or insecurity. As discussed above, based on the evidence found in developing countries, institutional determinants, such as trust in government and quality of public services, are expected to influence tax evasion [23].

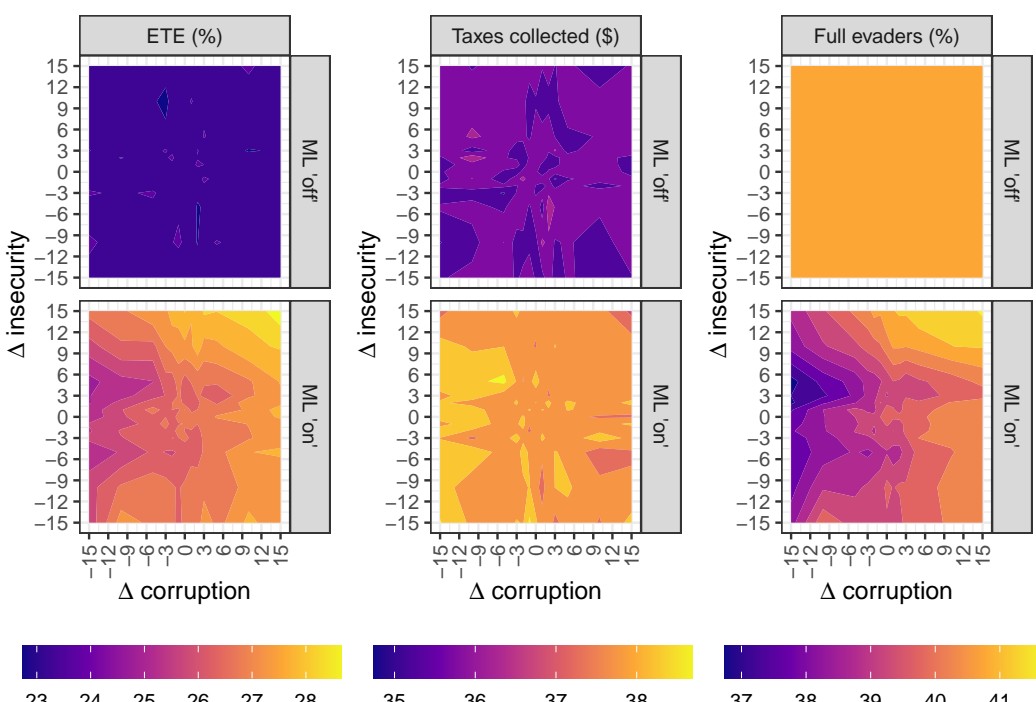

**Figure 9.** Percentage of ETE, monetary units of taxes collected, and percentage of full evaders in the system, when varying perception on corruption and insecurity, with and without machine learning. Purplish regions denote a small value of the output variable. A low value of ETE or full evaders was good, while, for taxes collected, a low value was bad.

However, it was appreciated that the results of the simulation with ML "off" failed to capture the changes in institutional determinants, mainly in two of the observed variables, the Extent of Tax Evasion, and the number of full evaders in the system. This meant that the analytical model gave greater importance to fiscal parameters (tax rate and penalty rate), and less to institutional parameters (variations in perceived corruption and insecurity).

Conversely, the model with ML "on" could reflect the variations of the institutional factors in a plausible way. For instance, if both the perception of corruption and insecurity increased, the Extension of Tax Evasion also increased. Paradoxically, at that point, there was also a high collection, generated by the greater number of penalties collected. It could also be noted that the perception of corruption played an important role in determining whether employers aligned themselves in the formal or informal sectors. Few evaders in the system and high collection were achieved if the perception of corruption decreased, even if the perception of insecurity increased slightly.

## 6. Discussion

Different applications of ML in ABM have been reported in the literature. We categorized them according to the moment learning was performed, resulting in online applications, i.e., learning during the simulation, and offline applications, i.e., learning before (a priori) or after (a posteriori) the simulation. Each case had different requirements. A priori learning used existing data about the application domain and might be performed before the ABM even existed to assist its design. A posteriori learning used data generated by the simulation to calibrate parameters and/or analyze the obtained results. On-line learning used data collected during the simulation to enhance the performance of the agents in a given task. For this, different kinds of supervised, evolutionary, and reinforcement learning algorithms were successfully applied.

Reproducible ABMs result from understandable and complete published descriptions. Standardized descriptions, such as those provided by the adoption of the ODD Protocol, contribute greatly in this sense. However, in its current state, the ODD Protocol can not accommodate the diversity of applications of ML in ABMs. Fortunately, the ODD Protocol covers other necessary elements to carry out such descriptions, simplifying its extension with the goal of documenting the diverse applications of ML identified in the ABM literature. For this, the proposed categorization of the ML applications in ABM was central, enabling the design questions guiding the extension to the learning component of the protocol, i.e., What is the purpose of learning? When is learning performed? What components of the ABM are affected by learning? How is learning computed?

The benefits of this extension to the ODD Protocol were illustrated with a case study. A full-featured ABM of the institutional factors on payroll tax evasion in Mexico. In this model, supervised ML was used to synthesize data from the tax environment and encapsulated itself as an agent that would be available to the rest of the agents in the model to make decisions about remaining in the formal or informal sector. The proposed extensions to the ODD protocol made it possible to communicate the ML workflow in the ABM in an orderly and transparent way, favoring its understanding and potential replication in other investigations.

Finally, analyzing the case study, it is important to highlight the contribution of ML to the understanding of the phenomenon. Payroll Tax in Mexico has a low rate, but a high penalty for amounts evaded. With these characteristics, analytical models fail to capture the differences produced by institutional determinants, which seem to have less importance in the mathematical formulation, but have been shown to, in fact, have an effect. The inclusion of ML in ABM allows agents to make decisions that better approximate the complete dynamics of the systems. Therefore, if the intention was to influence the decision of workers and help reduce the extent of tax evasion, a policy recommendation would be related to the enhancement of institutions and the improvement of the quality of public goods provided.

Future work includes applying the proposed extension of the ODD protocol to study cases considering other kinds of learning applications, e.g., the calibration of parameters and the analysis of the ABM results. With the objective of summarizing the basic concepts of Machine Learning and its applications in agent-based modeling in recent years, an analytical and systematic review is also contemplated. This wouold allow us to determine the prospects and challenges for agent-based modeling assisted by machine learning in the near future.

**Author Contributions:** Conceptualization, A.P.-L., A.G.-H., M.Q.-C. and N.C.-R.; Investigation, A.P.-L.; Writing—original draft, A.P.-L. and A.G.-H.; Writing—review & editing, A.P.-L., A.G.-H., M.Q.-C. and N.C.-R. All authors have read and agreed to the published version of the manuscript.

**Funding:** This research received no external funding.

**Informed Consent Statement:** Not applicable.

**Data Availability Statement:** Not applicable.

**Acknowledgments:** The first author thanks the National Council of Science and Technology (CONACYT, Mexico) for the scholarship 743662.

**Conflicts of Interest:** The authors declare no conflict of interest.

## Abbreviations

The following abbreviations are used in this manuscript:

| | |
|---|---|
| ML | Machine Learning |
| ABM | Agent-Based Model |
| ODD | Overview, Design concepts, Details |
| RL | Reinforcement Learning |
| ANN | Artificial Neural Network |
| GDP | Gross Domestic Product |
| ETE | Extent of Tax Evasion |

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
