# Peer review of "Agent-Based Models Assisted by Supervised Learning: A Proposal for Model Specification"

_electronics, doi:10.3390/electronics12030495_

Round 1
Reviewer 1 Report
The paper deals with the use of ABM along with a ML approach. I have the following observations:
- I think that the paper is vague and the authors should try to provide a title that is more appropriate to the paper and to the journal it is submitted to.
- Some key results should be provided in the abstract.
- Please do not use acronyms as titles of the chapters.
- Figure 1 resembles very much with the one in the ODD protocol - please remove it.
- Why haven't the authors used an ODD+D protocol?
- The beginning of section 3 seems like a literature review section. Please move it at the beginning of the paper.
- the values and types of the ABM attributes are not provided, please add this information, as in the current form the proposed approach cannot be reproduced. Please add any other details that are needed for ensuring the application reproductibility.
- The ABM are known from their real time graphical interface. Does the proposed ABM have one?
- Please better argue the values in table 3 - how have them been chosen?
- The validation process is not convincing. Please use some error indicators.
Author Response
Reviewer: 1
The paper deals with the use of ABM along with a ML approach. I have the following observations:
COMMENT # 1.1
I think that the paper is vague and the authors should try to provide a title that is more appropriate to the paper and to the journal it is submitted to.
Answer to comment 1.1
We are aware that Electronics is a journal dedicated to sciences of electronics and its applications. However, we are trying to submit our article to the special issue called Modeling and Simulation Methods: Recent Advances and Application, whose purpose is to present recent advances in methods and applications that focus on any aspect of system or process modeling and simulation. Since our article has a contribution to improve the integration and documentation of machine learning with agent-based modeling, we believe that the title is appropriate for this special issue and the journal.
COMMENT # 1.2
Some key results should be provided in the abstract.
Answer to comment 1.2
Key results were added in the abstract.
COMMENT # 1.3
Please do not use acronyms as titles of the chapters.
Answer to comment 1.3
Acronyms are no longer used in chapter titles.
COMMENT # 1.4
Figure 1 resembles very much with the one in the ODD protocol - please remove it.
Answer to comment 1.5
The Figure is used to clearly define the components that make up the protocol. The figure is not deleted, but the corresponding citation is made.
COMMENT # 1.5
Why haven't the authors used an ODD+D protocol?
Answer to comment 1.5
ODD+D was designed to describe ABM that includes human decision making. However, according to the literature review carried out, machine learning to support agent-based models encompasses a much broader set of tasks than those for agent decision-making. In general, assist in the design of the model itself, optimization of the behavior of the agents, and calibration/validation and analysis of the model results. Therefore, a contribution was necessary that considered all these different scenarios and that, to the best of our knowledge, do not yet exist.
COMMENT # 1.6
The beginning of section 3 seems like a literature review section. Please move it at the beginning of the paper.
Answer to comment 1.6
The first part of section 3, in which the case study was contextualized, was moved to section 1.
COMMENT # 1.7
the values and types of the ABM attributes are not provided, please add this information, as in the current form the proposed approach cannot be reproduced. Please add any other details that are needed for ensuring the application reproductibility.
Answer to comment 1.7
Types are provided in Table 1, and values were added in Table 2.
COMMENT # 1.8
The ABM are known from their real time graphical interface. Does the proposed ABM have one?
Answer to comment 1.8
Yes. At the beginning of section 4, a screenshot of the user interface is added, as well as an explanation of the interface elements.
COMMENT # 1.9
Please better argue the values in table 3 - how have them been chosen?
Answer to comment 1.9
The initial values used for the global parameters of the model are discussed in detail.
COMMENT # 1.10
The validation process is not convincing. Please use some error indicators.
Answer to comment 1.10
An error indicator is included to validate the results of the models with and without machine learning
Reviewer 2 Report
The proposed research focuses on agent-based modeling (ABM) and in particular the use of machine learning. The proposed work is detailed and thorough, however, there are a few updates that are needed before considering it for publication.
Authors need to present the ABM with ML in a form of the graphical model so a reader would be able to understand better.
There must be previous works in this domain as well, so they need to explain their results as well and then establish how your proposed model is better than the state of the art. Probably experimentation is required to do so to claim the superiority of the proposed modeling approach.
Author Response
Reviewer: 2
The proposed research focuses on agent-based modeling (ABM) and in particular the use of machine learning. The proposed work is detailed and thorough, however, there are a few updates that are needed before considering it for publication.
COMMENT # 2.1
Authors need to present the ABM with ML in a form of a graphical model so a reader would be able to understand better.
Answer to comment 2.1
A diagram was added in Section 2 to represent the different applications of ML in ABM. Also a diagram was included in Section 3 to describe the application of ML in the case of study.
COMMENT # 2.2
There must be previous works in this domain as well, so they need to explain their results as well and then establish how your proposed model is better than the state of the art. Probably experimentation is required to do so to claim the superiority of the proposed modeling approach.
Answer to comment 2.2
Since the main objective of the article is to propose a framework that allows the transparent description of the integration of ML in ABM, related work is included in the introduction to note why our contribution makes a relevant contribution to the literature.
Reviewer 3 Report
I would like to thank the authors for this interesting contribution. The methodology is largely clear, and the manuscript is carefully written as well. I have a few points to consider in the next version, please.
(1)
As a general remark, I see that great effort has been obviously done for the literature review. However, I would have liked it to be more selective with a deeper analysis and synthesis.
(2)
Some statements in the introduction need to be supported by some references. For example, in line 41-44.
(3)
My view is that the ideas proposed by the present study have been (largely) considered in the literature. For example, please see studies below. That said, I find that it is important to refer to such related contributions.
https://doi.org/10.1145/3200921.3200933
https://doi.org/10.1109/DS-RT47707.2019.8958703
(4)
I suggest including a diagram, in section 3 of the case study, to describe how the integration between simulation and ML was applied.
(5)
Please mention any libraries used to develop the models, and please cite their references.
Author Response
I would like to thank the authors for this interesting contribution. The methodology is largely clear, and the manuscript is carefully written as well. I have a few points to consider in the next version, please.
COMMENT # 3.1
As a general remark, I see that great effort has been obviously done for the literature review. However, I would have liked it to be more selective with a deeper analysis and synthesis.
Answer to comment 3.1
With the objective of summarizing the basic concepts of Machine Learning and its applications in agent-based modeling in recent years, an analytical and systematic review is contemplated as future work. This will allow us to determine the prospects and challenges for agent-based modeling assisted by machine learning in the near future.
COMMENT # 3.2
Some statements in the introduction need to be supported by some references. For example, in line 41-44.
Answer to comment 3.2
References are added in the introduction to support some arguments. Section 2 details those arguments with all the references.
COMMENT # 3.3
My view is that the ideas proposed by the present study have been (largely) considered in the literature. For example, please see studies below. That said, I find that it is important to refer to such related contributions.
https://doi.org/10.1145/3200921.3200933
https://doi.org/10.1109/DS-RT47707.2019.8958703
Answer to comment 3.3
Both contributions, as well as others existing in the literature, are included in the introduction as related work. Although those studies already consider ways of including ML in ABM, they do not give insights about how a researcher can carry out an implementation and documentation on the incorporation of both tools.
COMMENT # 3.4
I suggest including a diagram, in section 3 of the case study, to describe how the integration between simulation and ML was applied.
Answer to comment 3.4
A diagram was included in section 3 to describe the integration between ML and ABM.
COMMENT # 3.5
Please mention any libraries used to develop the models, and please cite their references.
Answer to comment 3.5
The libraries used to develop the models have already been added and cited.